# Collaborate to Adapt: Source-Free Graph Domain Adaptation via Bi-directional Adaptation

## ABSTRACT

Unsupervised graph domain adaptation has emerged as a practical solution to transfer knowledge from a label-rich source graph to a completely unlabelled target graph, when there is a scarcity of labels in target graph. However, most of existing methods require a labelled source graph to provide supervision signals, which might not be accessible in the real-world scenarios due to regulations and privacy concerns. In this paper, we explore the scenario of source-free unsupervised graph domain adaptation, which tries to address the domain adaptation problem without accessing the labelled source graph. Specifically, we present a novel paradigm called GraphCTA, which performs model adaptation and graph adaptation collaboratively through a series of procedures: (1) conduct model adaptation based on node's neighborhood predictions in target graph considering both local and global information; (2) perform graph adaptation by updating graph structure and node attributes via neighborhood constrastive learning; and (3) the updated graph serves as an input to facilitate the subsequent iteration of model adaptation, thereby establishing a collaborative loop between model adaptation and graph adaptation. Comprehensive experiments are conducted on various public datasets including transaction, social, and citation graphs. The experimental results demonstrate that our proposed model outperforms recent source free baselines by large margins. Our source code and datasets are available at https://anonymous.4open.science/r/GraphCTA-code.

## KEYWORDS

Graph Representation Learning, Graph Domain Adaptation

**ACM Reference Format:**
Anonymous Author(s). 2024. Collaborate to Adapt: Source-Free Graph Domain Adaptation via Bi-directional Adaptation. In *Proceedings of the ACM Web Conference 2024 (WWW '24), May 13 – May 17, 2024, Singapore.* ACM, New York, NY, USA, 16 pages. https://doi.org/10.1145/nnnnnnn.nnnnnnn

## 1 INTRODUCTION

The Web is a complex network of interconnected entities, which can be effectively represented using graph structures. Graph techniques have demonstrated impressive performance in various web applications such as online article classification [22, 49], web-scale recommendation systems [10, 59], and anomaly detection [9, 45], etc. Undoubtedly, Graph Neural Networks (GNNs) have emerged as

a powerful tool when handling graph-structured data across a broad range of applications. Despite its success, the performance improvement often comes at the cost of utilizing sufficient high-quality labels. Unfortunately, obtaining enough labels for graph-structured data could be a laborious and time-consuming task. For instance, annotating the properties of molecular graphs requires expertise in chemical domains and rigorous laboratory analysis [19]. To alleviate the burden of laborious data annotations, Domain Adaptation (DA) presents an attractive option to transfer the knowledge learned from the labelled source domain to the unlabelled target domain. However, GNN models trained on source domains typically experience significant performance degradation when directly applied to target domains, due to the issue of domain shift [3, 63, 69]. Considerable endeavors have been dedicated to learning domain invariant representations, thereby enhancing the model's ability to generalize across different domains.

Recently, two mainstream strategies have been explored for unsupervised graph domain adaptation. One research line is to explicitly minimize the distribution discrepancy between the source and target representations [42, 52, 60]. How to define an appropriate discrepancy metric plays an important role in this kind of methods. Two commonly adopted measures to match cross-domain representations are the maximum mean discrepancy [30] and central moment discrepancy [61]. Another direction is to learn domain invariant representations via adversarial training [7, 41, 53], which achieves implicit representation alignment through a domain discriminator. Its flexibility of not requiring a predefined metric has made it gain increased popularity. Nonetheless, these joint learning approaches require the authorization to access the source data, which poses great challenges regarding data privacy and intellectual concerns. In most practical scenarios, the only accessible resources for domain adaptation are unlabelled target data and a model trained on source data, which is named source-free unsupervised domain adaptation.

Let's imagine a situation where a financial institution operates globally, processing a large number of transactions from domestic and overseas sources. Given the sensitivity of customer information involved in these transactions, privacy regulations restrict the institution's access to transaction data across different countries, such as the European Union General Data Protection Regulation (EU GDPR) and Singapore's Personal Data Protection Act (PDPA), etc. By utilizing source-free graph adaptation, the financial institution can adapt fraud detection models that have been trained on the domestic transaction graph to be applicable to overseas graphs, while respecting privacy regulations that limit the sharing of transaction data across countries. In contrast, the aforementioned source-need domain adaptation models are not applicable in this scenario due to significant privacy concerns associated with accessing and utilizing the labelled source data.

While source-free unsupervised domain adaptation has been extensively studied for image and text data [6, 24, 25, 27, 67], there has been limited investigation of source-free adaptation techniques for the non-iid graph-structured data. It involves two primary challenges in this scenario: *(1) How can adaptation be achieved without accessing the labelled source graph? (2) How to mitigate distribution shifts induced by node features as well as graph structures?* For instance, in the context of citation networks, when the topic of a research filed gains increasing popularity, such as the rise of artificial intelligence and large language models, the node features (i.e., the contents of the papers) and graph structures (i.e., the citation relationships between the papers) might undergo significant changes over the time. The complex interactions among different nodes present great challenges when attempting to adapt the GNN model trained on an earlier version of the citation network (e.g., before 2010) to a more recent version (e.g., after 2010). Meanwhile, without graph labels for supervision, the patterns learned from the source graph may not be suitable for the target graph, which suffers source hypothesis bias and results in false predictions in the target graph. One recent work SOGA [34] performs source-free domain adaptation on graphs, but it only focuses on the local neighbor similarity within the target graph, overlooking the global information and the inherent graph discrepancy. Hence, it is necessary to design source-free graph domain adaptation techniques that specifically tackle the challenges posed by graph-structured data, while overcoming the limitations of existing approaches.

To address the aforementioned challenges, we propose a novel framework abbreviated as GraphCTA (**C**ollaborate **T**o **A**dapt), which achieves source-free graph domain adaptation via collaboratively bi-directional adaptations from the perspectives of GNN model and graph data. More specifically, to learn node representations that are invariant to arbitrary unknown distribution shifts, GraphCTA generates node representations with selected node neighborhoods and complemented node features. Then, we perform *model-view* adaptation according to its local neighborhood predictions and the global class prototypes. Memory banks are used to store all target representations and their corresponding predictions through momentum updating [17], which generates robust class prototypes and ensures consistent predictions during the training stage. To filter out noisy neighbors and complement node features, we further propose to conduct *graph-view* adaptation based on the model's predictions and the information stored in the memory banks. Particularly, we derive pseudo labels from high-confidence target samples and utilize neighborhood contrastive learning to guide the graph adaptation procedure. By using the updated graph as input, we enable the next round of model adaptation and establish a collaborative loop between the model and the graph adaptation. *Theoretical analysis shows that adapting model and graph data collaboratively can reduce the upper bound of target domain prediction error in Appendix A.* We comprehensively evaluate GraphCTA on multiple benchmarks, and the experimental results demonstrate the effectiveness of our proposed approach, which can even outperform source-need baselines in various scenarios.

To summarize, the main contributions are as follows:

- We investigate the problem of source-free unsupervised graph domain adaptation without access to labelled source

graphs during the target adaptation, which is more practical in real-world scenarios and less explored in the literature of graph neural networks.
- To the best of our knowledge, we are the first to perform model adaptation and graph adaptation collaboratively, which is model-agnostic and can be applied to numerous GNN architectures.
- Extensive experimental results show the effectiveness of our method, with GraphCTA outperforming the SOTA baselines by an average of 2.14% across multiple settings.

## 2 RELATED WORK

**Graph Neural Networks.** GNNs have led to significant advancements in graph-related tasks, which incorporate graph structural information via message passing mechanism. Various models have been proposed to enhance their performance and extend their applications. In general, they can be classified into two categories: spectral based and spatial based methods. For spectral approaches, the graph convolution is performed on the spectrum of graph Laplacian. Among them, ChebNet [8] leverages Chebyshev polynomials to approximate graph filters that are localized up to K orders. ARMA [4] uses auto-regressive moving average filter to capture global graph structure. GCN [22] simplifies ChebNet by truncating the Chebyshev polynomial to the first-order, leading to high efficiency. As for spatial methods, the graph convolution is designed to directly aggregate the neighborhood information of each node. For instance, GraphSAGE [16] proposes various aggregator architectures (i.e., mean, LSTM) to aggregate its local neighborhood. GAT [49] employs an attention mechanism to adaptively aggregate node's neighborhood representations. SGC [51] further simplifies the graph convolution by eliminating nonlinearities and collapsing weight matrices between consecutive layers. More detailed introduction can be found in various comprehensive surveys on graph neural networks [54, 68].

**Domain Adaptation.** Domain adaptation aims to enhance the model's ability to generalize across domains, which transfers the knowledge learned from a labelled source domain to unlabelled target domain. The model's performance may suffer from a significant degradation in target domain due to the domain shifts. To address this challenge, many approaches are proposed to learn domain invariant representations in the field of computer vision and natural language processing [50, 58]. Among them, [30, 32, 33] try to explicitly align source and target feature distributions via minimizing maximum mean discrepancy. Similarly, [44, 61, 62] utilize central moment discrepancy to match high order statistics extracted by neural networks. Instead of directly aligning feature distributions, [18, 31, 47] employ adversarial training strategy to generate indistinguishable source and target representations, where domain invariance is formulated as a binary domain classification problem. All the above mentioned methods assume both source and target data are available during the adaptation procedure, which may not be feasible in real-world scenarios due to privacy concerns. Some recent works [6, 24, 25, 27] investigate source-free domain adaptation, where only well-trained source model and unlabelled target domain data are accessible. Specifically, SHOT [25] utilizes pseudo

labeling strategy associated with entropy minimization and information maximization to optimize the model on target domain. NRC [57] encourages consistency via neighborhood clustering, where reciprocal neighbors and expanded neighborhoods are incorporated to capture their local structure. JMDS [24] robustly learns with pseudo-labels by assigning different confidence scores to the target samples. However, these methods are specifically designed for independent and identically distributed data, which may not be appropriate for non-iid graph-structured data.

**Graph Domain Adaptation.** Graph provides a natural way to represent the intricate interactions among different entities, which leads to non-trivial challenges for domain adaptation tasks because of its non-iid properties. There have been some recent efforts that focus on unsupervised graph domain adaptation [7, 34, 52, 53, 60]. Particularly, [42] follows the idea of feature alignments in feature space and utilizes maximum mean discrepancy to yield domain invariant node representations. UDAGCN [53], ACDNE [41] and AdaGCN [7] adopt the techniques of adversarial training to mitigate the distribution divergence, where the difference lies at how they generate effective node representations. ASN [64] disentangles the knowledge into domain-private and domain-shared information, then adversarial loss is adopted to minimize the domain discrepancy. GRADE [52] employs graph subtree discrepancy to quantify the distribution shift between source and target graphs. SpecReg [60] proposes theory-grounded algorithms for graph domain adaptation via spectral regularization. Likewise, the aforementioned methods rely heavily on the supervision signals provided by the labelled source graph, which is usually inaccessible due to privacy preserving policies. Lately, SOGA [34] studies source-free unsupervised graph domain adaptation through preserving the consistency of structural proximity on the target graph. Nevertheless, it follows existing works that perform model adaptation, neglecting the fact that the domain shift is caused by the target graph's property. In contrast, our proposed GraphCTA conducts model adaptation and graph adaptation collaboratively to address this problem.

## 3 THE PROPOSED GRAPHCTA

### 3.1 Preliminary and Problem Definition

For source-free unsupervised graph domain adaptation, we are provided with a source pre-trained GNN model and an unlabelled target graph $\mathcal{G} = (\mathcal{V}, \mathcal{E}, \mathbf{X})$, where $\mathcal{V}$ and $\mathcal{E}$ denote the node and edge sets, respectively. The edge connections are represented as adjacent matrix $\mathbf{A} \in \mathbb{R}^{n \times n}$ and $\mathbf{A}_{i,j} = 1$ if $v_i$ connects to $v_j$, while the node feature matrix $\mathbf{X} \in \mathbb{R}^{n \times d}$ specifies the features of the nodes. Here, $n$ indicates the number of nodes and $d$ is the dimension of the node features. In this paper, we mainly focus on a $C$-class node classification task in the closed-set setting, where the labelled source graph and unlabelled target graph share the same label space. We further partition the GNN model into two components: the feature extractor $f_\theta(\cdot)$ that maps graph $\mathcal{G}$ into node representation space $\mathbb{R}^{n \times h}$ and the classifier $g_\phi(\cdot)$ which projects node representations into prediction space $\mathbb{R}^{n \times C}$. Given the aforementioned notations, we can provide a formal definition of our problem as follows:

*Definition 3.1 (Source-Free Unsupervised Graph Domain Adaptation). Given a well-trained source GNN model $\kappa = f_\theta \circ g_\phi$ and an*

unlabelled target graph $\mathcal{G}$ under the domain shift, our goal is to adapt the source pre-trained model to perform effectively on the target graph without any supervision, where the GNN architecture and domain shift can be arbitrary.

To adapt the given source pre-trained model, we address the aforementioned challenges by optimizing the GNN model as well as the target graph data to reduce the gap between source and target domains. Figure 1 provides an overall view of our proposed GraphCTA, which consists of two key components: a *model adaptation* module and a *graph adaptation* module. In the subsequent sections, we will elaborate the details of different components.

### 3.2 Model Adaptation with Local-Global Consistency

**Domain-shift Invariant Node Representation Learning.** To mitigate the source hypothesis bias in the target graph, we optimize the source pre-trained GNN model's parameters to generate domain-shift invariant node representations. As GNN models mainly involve propagating and aggregating information from its structural neighborhood, we propose to complement node features and adaptively select node neighborhood when modeling their interactions. Specifically, let $\mathbf{Z} \in \mathbb{R}^{n \times h}$ denote the node representations extracted by $f_\theta(\cdot)$, which is updated as follows:

$$\mathbf{z}_i^l = \text{UPDATE}^l(\mathbf{z}_i^{l-1}, \text{AGG}^l(\{\mathbf{z}_u^{l-1} | u \in \psi(\mathbf{A}_i)\})), \quad (1)$$

where $\mathbf{z}_i^l$ is node $v_i$'s representation at layer $l$ with $\mathbf{z}_i^0 = \delta(\mathbf{x}_i)$. $\mathbf{A}_i$ represents $v_i$'s neighborhood. $\delta(\cdot)$ and $\psi(\cdot)$ indicate the node feature complementary and neighborhood selection functions, which will be introduced in Section 3.3. $\text{AGG}(\cdot)$ refers to an aggregation function that maps a collection of neighborhood representations to an aggregated representation. $\text{UPDATE}(\cdot)$ combines the node's previous and aggregated representations. For readability, we will omit the superscript $l$ and use $\mathbf{Z}$ to denote the node representations in the following sections.

**Neighborhood-aware Pseudo Labelling.** Since the target representations extracted from the source pre-trained model already form semantic clusters, we propose to achieve model adaptation by encouraging neighborhood prediction consistency. The pseudo labels are generated by aggregating the predicted neighborhood class distributions. However, the local neighborhood could produce noisy supervision signal due to the domain-shift. We further assign a confidence score to each target sample according to the semantic similarities with global class prototypes, which mitigates the potential negative influence introduced by its local neighbors. To generate stable class prototypes and prediction distributions, we build target representation memory bank $\mathcal{F} = [\mathbf{z}_1^m, \mathbf{z}_2^m, \cdots, \mathbf{z}_n^m]$ and predicted distribution memory bank $\mathcal{P} = [\mathbf{p}_1^m, \mathbf{p}_2^m, \cdots, \mathbf{p}_n^m]$, which are updated via a momentum strategy during the training procedure:

$$\mathbf{z}_i^m = (1 - \gamma)\mathbf{z}_i^m + \gamma\mathbf{z}_i, \quad (2)$$

where $\gamma$ is the momentum coefficient. For memory bank $\mathcal{P}$, we first sharpen the output predictions $\mathbf{p}_i = \mathbf{p}_i^2 / \sum_{j=1}^n \mathbf{p}_j^2$ to reduce the ambiguity in the predictions. $\mathbf{p}_i = g_\phi(\mathbf{z}_i) \in \mathbb{R}^C$ represents the predicted class distribution. Then, the values stored in the memory bank $\mathcal{P}$ are updated following a similar procedure in Eq. (2).

**Figure 1: The overall architecture of our proposed GraphCTA framework, which is composed of model adaptation and graph adaptation.**

With the neighborhood information, we compute the one-hot pseudo label distribution of node $v_i$ as follows:

$$\hat{\mathbf{p}}_i = \mathbb{1}\left[\arg\max_c \left(\frac{1}{|\mathcal{N}(i)|} \sum_{j \in \mathcal{N}(i)} \mathbf{p}_j^m\right)\right], \tag{3}$$

where $\mathbb{1}[\cdot]$ is the one-hot function that encodes pseudo labels. $\mathcal{N}(i) = \{v_j | j \in \psi(\mathbf{A}_i)\}$ denotes the selected node neighborhood of node $v_i$ and $\mathbf{p}_j^m$ is the predicted distribution stored in the memory bank $\mathcal{P}$. As the pseudo label depends heavily on the graph's local structure and does not take the global contextual information into consideration, it could jeopardize the training process and result in erroneous classifications. Thus, we include global class-wise prototypes to weigh the generated pseudo labels. The prototypes provide an estimation of the centroid for each class, which can be calculated as follows:

$$\boldsymbol{\mu}_c^m = \frac{\sum_{i=1}^n \mathbb{I}(\hat{\mathbf{p}}_{i,c} = 1) \cdot \mathbf{z}_i^m}{\sum_{i=1}^n \mathbb{I}(\hat{\mathbf{p}}_{i,c} = 1)}, \tag{4}$$

where $\mathbb{I}(\cdot)$ is the indicator function. $\mathbf{z}_i^m$ represents the node representation stored in the memory bank $\mathcal{F}$. Then, we define the confidence score for each sample as the semantic similarity between the target representation and its corresponding pseudo class prototype calculated from memory bank. Here, we choose cosine similarity for simplicity:

$$\text{sim}(\mathbf{z}_i, \boldsymbol{\mu}_c^m) = \frac{\mathbf{z}_i^\top \boldsymbol{\mu}_c^m}{\|\mathbf{z}_i\|_2 \cdot \|\boldsymbol{\mu}_c^m\|_2}, \tag{5}$$

where it gives high confidence scores whose representations are consistent with class-wise prototypes.

**Local-Global Consistency Optimization.** Afterwards, we fine-tune the model's parameters by optimizing the weighted cross-entropy loss between the pseudo label distribution and the predicted class distribution:

$$\mathcal{L}_{\text{CE}} = -\frac{1}{n} \sum_{i=1}^n \sum_{c=1}^C \text{sim}(\mathbf{z}_i, \boldsymbol{\mu}_c^m) \cdot \hat{\mathbf{p}}_{i,c} \log(\mathbf{p}_{i,c}). \tag{6}$$

Additionally, we further consider instance-prototype alignment inspired by recent contrastive learning [5, 17, 43] to regularize the

learned representations, which maximizes the similarity between the node representation and its corresponding prototype. The remaining $C - 1$ prototypes and $n - 1$ instance representations are regarded as negative pairs that are pushed apart in the latent space. The contrstive loss can be formulated as the following InfoNCE loss [5]:

$$\mathcal{L}_{\text{CO}} = -\frac{1}{n} \sum_{i=1}^n \log \frac{\exp(\text{sim}(\mathbf{z}_i, \boldsymbol{\mu}_c^m)/\tau)}{\begin{cases} \sum_{j=1}^C \mathbb{I}(j \neq c)\exp(\text{sim}(\mathbf{z}_i, \boldsymbol{\mu}_j^m)/\tau) \\ + \sum_{k=1}^n \mathbb{I}(k \neq i)\exp(\text{sim}(\mathbf{z}_i, \mathbf{z}_k)/\tau) \end{cases}}, \tag{7}$$

where the temperature $\tau$ is a hyper-parameter. Note that the contrastive loss is also able to model the local and global information simultaneously. By integrating these two losses, we can obtain the final objective for model adaptation as follows:

$$\mathcal{L}_{\text{M}} = (1 - \lambda)\mathcal{L}_{\text{CE}} + \lambda\mathcal{L}_{\text{CO}}, \tag{8}$$

where $\lambda$ is the trade-off parameter.

### 3.3 Graph Adaptation with Self-training

As we have discussed earlier, the performance degradation in target graph can be attributed to the presence of source hypothesis bias and domain shift. Although the model adaptation module can help alleviate the source hypothesis bias to some extent, the underlying domain shift originates from the characteristics of the input graph data. However, most existing approaches mainly focus on designing model adaptation techniques [34, 52, 53, 60], neglecting the fact that the domain shift is aroused from the target graph itself. Therefore, we propose to perform graph adaptation by refining the graph data to make them more compatible between the domains.

**Node Feature and Neighborhood Refinement.** Specifically, we introduce two simple transformation functions: $\mathbf{X}' = \sigma(\mathbf{X})$ which produces new node features by adding or masking values in $\mathbf{X}$, and $\mathbf{A}' = \psi(\mathbf{A})$ which generates new adjacent matrix via connecting or deleting edges in $\mathbf{A}$. The goal of graph adaptation module is to find optimal functions that can reduce the domain shift. However, it is a non-trivial task due to the absence of supervision and the unavailability of source graph. While a variety of choices

are available to alter the graph data, for instance, the graph structure learning mechanisms [11, 20, 29, 65], we adopt two extremely simple and straightforward policies below. More choices are discussed in ablation study Section 4.3.2.

Given node feature matrix $\mathbf{X}$, we formulate node feature transformation as $\mathbf{X}' = \sigma(\mathbf{X}) = \mathbf{X} + \Delta\mathbf{X}$, which utilizes an additive function to complement node features. $\Delta\mathbf{X} \in \mathbb{R}^{n \times d}$ are continuous free parameters and provide high flexibility. This approach enables either the masking of node features to zeros or the modification of these features to alternate values. Similarly, we model the graph structure as $\mathbf{A}' = \psi(\mathbf{A}) = \mathbf{A} \oplus \Delta\mathbf{A}$, where $\Delta\mathbf{A} \in \mathbb{R}^{n \times n}$ represents a binary matrix to refine the node's neighborhood and $\oplus$ means the element-wise exclusive OR operation (i.e, XOR). That's to say, if the elements in $\mathbf{A}$ and $\Delta\mathbf{A}$ are both 1, the XOR operation returns 0 and results in edge deletion. If elements in $\mathbf{A}$ and $\Delta\mathbf{A}$ are 0 and 1 respectively, it leads to the edge additions. To prevent significant deviations from the original graph structure, we impose a constraint on the maximum number of modified entries in the adjacency matrix to be less than a predetermined budget $\mathcal{B}$, i.e., $\sum \Delta\mathbf{A} \leq \mathcal{B}$, which reduces the search space and is computation efficient.

**Self-Training with Neighborhood Contrastive Learning.** In order to optimize the free-parameters $\Delta\mathbf{X}$ and $\Delta\mathbf{A}$, we propose to employ a self-training mechanism to guide the graph adaptation procedure, since the ground-truth labels are not available under this setting. In particular, we first identify a set of reliable sample pairs via its prediction confidence as follows:

$$\mathcal{D} = \{(v_i, \hat{y}_i) | \hat{y}_i = \arg\max_c \mathbf{p}_{i,c} \wedge \max(\mathbf{p}_i) > \omega, v_i \in \mathcal{V}\}, \quad (9)$$

where a predefined threshold $\omega$ is utilized to select the high confidence target samples (i.e., $\omega = 0.9$) and $\hat{y}_i$ denotes its corresponding pseudo label. Different from model adaptation module that leverages local neighborhood to construct pseudo labels, here we solely rely on the sample's own prediction since our goal is to refine the graph structure. In this scenario, its structural neighborhood cannot be regarded as a reliable supervision signal. To exploit the intrinsic local structure in the representation space, we further incorporate neighborhood constrastive learning to push similar samples closer and dissimilar samples apart. Then, the positive samples are generated by extracting $K$-nearest neighbors in memory bank $\mathcal{F}$ via cosine similarity as follows:

$$\chi_i = \{\mathbf{z}_j^m | \arg\text{topk}(\text{sim}(\mathbf{z}_i, \mathbf{z}_j^m)), \mathbf{z}_j^m \in \mathcal{F}\}, \quad (10)$$

where $\text{topk}(\cdot)$ is a function returning the most similar $K$ samples. Next, we use those samples whose predicted labels are different from $\mathbf{p}_i$ to form negative samples:

$$\Psi_i = \{\mathbf{z}_j^m | \arg\max_c \mathbf{p}_i \neq \arg\max_c \mathbf{p}_j^m, \mathbf{z}_j^m \in \mathcal{F} \wedge \mathbf{z}_j^m \notin \chi_i\}, \quad (11)$$

where $\mathbf{p}_i^m$ and $\mathbf{z}_i^m$ are from memory banks. Through this way, the knowledge gained from the model adaptation module can facilitate the learning process of graph adaptation. To sum up, the overall loss function for graph adaptation is:

$$\mathcal{L}_G = -\frac{1}{|\mathcal{D}|} \sum_{i \in \mathcal{D}} \log(\mathbf{p}_{i,\hat{y}_i}) - \alpha \sum_{i=1}^n \sum_{j \in \chi_i} \text{sim}(\mathbf{z}_i, \mathbf{z}_j^m)$$
$$+ \beta \sum_{i=1}^n \sum_{k \in \Psi_i} \text{sim}(\mathbf{z}_i, \mathbf{z}_k^m), \quad (12)$$

where $\alpha$ and $\beta$ are hyper-parameters to balance the cross-entropy and the neighborhood contrastive learning loss. Since $\Delta\mathbf{A}$ is binary

and constrained, we relax the binary space to a continuous space $[0, 1]^{n \times n}$ and employ projected gradient descent (PGD) [14, 55] for updating $\Delta\mathbf{A}$. *More details are given in Appendix B.*

## 3.4 The Training Procedure

We employ an alternative training strategy to iteratively update these two collaborative components, i.e., model adaptation module and graph adaptation module. Specifically, in each training epoch, we first update the parameters of graph adaptation module $\kappa$ to minimize $\mathcal{L}_M$ while keeping $\Delta\mathbf{X}$ and $\Delta\mathbf{A}$ fixed. Then, $\Delta\mathbf{X}$ and $\Delta\mathbf{A}$ are updated to optimize $\mathcal{L}_G$ while keeping model $\kappa$ fixed. To facilitate the understanding of our training procedure, *we provide a detailed description of the whole process in Algorithm 1 at Appendix C*, which outlines the step-by-step process we have adopted to update the collaborative components.

## 3.5 Complexity Analysis

Assume that we have a graph consisting of $n$ nodes and $e$ edges, the node representation dimension is set as $h$ and the number of graph neural network layers is $L$. Then, the time complexity of feature encoder is $O(Lnh^2 + Leh)$. In model adaptation, generating pseudo labels has the time complexity of $O(eC + nC)$, where $C$ is the number of class. The complexity of calculating prototypes and confidence scores is $O(nh + nhC)$. The contrastive loss has the time complexity of $O(n^2h)$. In graph adaptation, node feature transformation has the complexity of $O(nd)$, where $d$ is the node feature dimension. The time complexity of structure refinement is constrained to $O(e)$ and the neighborhood contrastive learning has the time complexity of $O(n^2h)$. When employing batch updating, the time complexity of contrastive learning module can be reduced to $O(B^2h)$, where $B$ represents the batch size. If we further take the localization properties of the graph into consideration, the time complexity for computing K-nearest neighbors in memory bank can be reduced to $O(Tnd)$, where $T$ is average node neighbors within node's $t$-hop. Thus, the overall time complexity of our proposed GraphCTA is within the same scope of vanilla GNN.

## 4 EXPERIMENTS

## 4.1 Experimental Settings

**Datasets.** Our proposed GraphCTA is evaluated on three public datasets with node classification task, and a summary of their statistics is provided in Table 1, which includes three types of distribution shifts. Among them, **Elliptic**[1] [36] is a temporal bitcoin transaction graph containing a sequence of graph snapshots, where each edge represents a payment flow and each node is labelled as licit, illicit or unknown. Then, we construct three domains by grouping the first 10 start snapshots as Elliptic-S, the middle 10 snapshots as Elliptic-M and the last 10 end snapshots as Elliptic-E according to their chronological order. In this scenario, the model needs to handle the temporal shifts, since the distributions for node features and edges are highly correlated with time. **Twitch**[2] [38] consists of several social networks collected from different regions, in which the nodes are users and the edges denote their friendships. We choose three

---

[1]https://www.kaggle.com/datasets/ellipticco/elliptic-data-set
[2]https://github.com/benedekrozemberczki/datasets#twitch-social-networks

**Table 1: Dataset Statistics.**

| Category | Dataset | Distribution Shift | #Nodes | #Edges | #Features | #Classes |
|---|---|---|---|---|---|---|
| | Elliptic-S | | 58,097 | 71,732 | | |
| Transaction | Elliptic-M | Temporal Level | 34,333 | 38,171 | 165 | 3 |
| | Elliptic-E | | 46,647 | 53,491 | | |
| | Twitch-DE | | 9,498 | 153,138 | | |
| Social | Twitch-EN | Domain Level | 7,126 | 35,324 | 3,170 | 2 |
| | Twitch-FR | | 6,549 | 112,666 | | |
| | ACMv9 | | 9,360 | 15,556 | | |
| Citation | Citationv1 | Temporal & Domain | 8,935 | 15,098 | 6,775 | 5 |
| | DBLPv7 | | 5,484 | 8,117 | | |

largest graphs to perform adaptation, i.e., Germany (GE), England (EN) and France (FR). The node features are extracted based on the games played and liked by users, their locations and streaming habits, etc. Each user is binary-labelled, indicating whether they use explicit language. **Citation**[3] [64] involves three citation datasets provided by ArnetMiner [46] extracted from different sources and time periods. Specifically, ACMv9 (A), Citationv1 (C), DBLPv7 (D) are derived from ACM (between years 2000 and 2010), Microsoft Academic Graph (before the year 2008) and DBLP (between years 2004 and 2008), respectively. Then, each paper is classified into five categories (i.e, DB, AI, CV, IS and Networking) according to its research topic. The distribution shifts are aroused from both temporal and domain levels. *More detailed information can be found in Appendix D.1.*

**Baselines.** We compare GraphCTA with baselines including *no-adaptation*, *source-need* and *source-free* domain adaptation approaches. For no-adaptation methods, the model is first trained on the source graph, and then directly evaluated on the target graph without any adaptation operations. In contrast, source-need methods optimize the model with both source and target graphs through implicit or explicit metrics to align their distributions. *We refer readers to Appendix D.2 for more detailed description.* Here, we briefly introduce some of the most related SOTA source-free models. As pioneers in exploring the novel and crucial setting of source-free graph domain adaptation, we conduct a comprehensive comparison with baselines from both computer vision and graph domains. Among them, SHOT [25] and its extension SHOT++ [27] employ entropy minimization and information maximization to perform class-wise adaptation. BNM [6] achieves prediction discriminability and diversity via nuclear norm maximization. ATDOC [26] and NRC [57] exploit local neighborhood structure for ensuring label consistency. DaC [67] partitions the target data into source-like and target-specific samples to perform domain adaptation. JMDS [24] assign confidence score to each target sample for robust adaptation learning. GTRANS [20] performs graph transformation at test time to enhance the model's performance. SOGA[4] [34] maximizes the mutual information between the target graph and the model's output to preserve the structural proximity.

**Implementation Details.** Similar to previous works [34, 53], we randomly split each source graph into 80% as training set, 10% as validation set and the remaining 10% as test set. The source GNN model is first supervised and pre-trained on the training set, followed by

tuning its hyper-parameters on the validation set. The test set in source graph serves as a sanity check to ensure a well-pretrained GNN model, and its final performance is evaluated on the whole target graph. We utilize the source codes provided by the authors and adopt the same GNN backbone with same number of layers. The node representation dimension is set as 128 for all the baselines. Our proposed GraphCTA is implemented with Pytorch Geometric[5] [12] and optimized with Adam optimizer [21]. The optimal learning rate and weight decay are searched in $\{0.1, 0.01, 1e^{-3}, 1e^{-4}, 5e^{-4}\}$. The smoothing parameter $\gamma$ in memory banks is fixed as 0.9 by default. Temperature $\tau$ and the number of $K$-nearest neighbors are set as 0.2 and 5, respectively. Trade-off hyper-parameters $\lambda, \alpha, \beta$ are searched in the range of $[0, 1]$. *Additional details for reproducibility are provided in Appendix G.*

## 4.2 Results and Analyses

Table 2 shows the node classification performance across 9 adaptation tasks from 3 datasets. We repeat the experiments 5 times with different seeds and then report their mean accuracy with standard deviation. *The overall experimental results are reported in Appendix E.* As can be seen from Table 2, the upper parts present the results of source-need approaches that *have access to the labelled source graph during adaptation.* The middle and lower parts show the results for no-adaptation and source-free methods that does not utilize the labelled source graph. In summary, our proposed GraphCTA is on par with source-need algorithms and even surpasses them in certain scenarios (i.e., DE→FR and C→D). Particularly, our method consistently achieves state-of-the-art performance on all tasks under the source-free setting. *It outperforms the strongest source-free baseline by a large margin (2.14% absolute improvements on average).* We note that the unsupervised method GAE demonstrates comparable performance on several specific tasks. However, its performance exhibits significant variation depending on the characteristics of the input graph, and thus fails to achieve consistent results in the context of domain adaptation. *Additionally, it can be observed that negative transfer occasionally occurs in these models, which is consistent with previous works' findings.* For instance, some source-need baselines (e.g., AdaGCN) and source-free methods (e.g., SHOT) perform worse than vanilla GCN without adaptation under the scenario of M→E. Moreover, it is more commonly observed in the source-free setting than in the source-need setting, primarily due to the lack of available source graph. Finally, our proposed GraphCTA can adapt to different types of graphs and adaptation

---

[3]https://github.com/yuntaodu/ASN/tree/main/data
[4]Their citation datasets are similar but distinct from ours. Please refer to Appendix H.

[5]https://pytorch-geometric.readthedocs.io/en/latest/

**Table 2: Average node classification performance in terms of accuracy (%). OOM means out-of-memory. We use blue to denote the best performance in source-need methods and bold indicates the best performance among source-free approaches.**

| | Methods | S→M | S→E | M→E | DE→EN | DE→FR | EN→FR | A→D | C→D | C→A |
|---|---|---|---|---|---|---|---|---|---|---|
| Source-Need | UDAGCN [53] | 81.12±0.04 | 73.91±0.64 | 77.22±0.16 | 59.74±0.21 | 56.61±0.39 | 56.94±0.70 | 66.95±0.45 | 71.77±1.09 | 66.80±0.23 |
| | TPN [35] | 82.06±0.19 | 76.59±0.70 | 79.17±0.33 | 54.42±0.19 | 43.43±0.99 | 38.93±0.28 | 69.78±0.69 | 74.65±0.74 | 67.93±0.34 |
| | AdaGCN [7] | 77.49±1.07 | 76.02±0.54 | 73.57±2.03 | 54.69±0.50 | 37.62±0.51 | 40.45±0.24 | 75.04±0.49 | 75.59±0.71 | 71.67±0.91 |
| | ASN [64] | OOM | OOM | OOM | 55.45±0.11 | 47.20±0.84 | 40.29±0.55 | 73.80±0.40 | 76.36±0.33 | 72.74±0.49 |
| | ACDNE [41] | 86.27±1.23 | 80.66±1.11 | 81.37±1.20 | 58.08±0.97 | 54.01±0.30 | 57.15±0.61 | 76.24±0.53 | 77.21±0.23 | 73.59±0.34 |
| | GRADE [52] | 79.77±0.01 | 74.41±0.03 | 78.84±0.06 | 56.40±0.05 | 46.83±0.07 | 51.17±0.62 | 68.22±0.37 | 73.95±0.49 | 69.55±0.78 |
| | SpecReg [60] | 80.90±0.06 | 75.89±0.06 | 77.65±0.02 | 56.43±0.11 | 63.20±0.03 | 63.21±0.04 | 75.93±0.89 | 75.74±1.15 | 72.04±0.63 |
| No-Adaptation | DeepWalk [37] | 75.52±0.01 | 75.98±0.02 | 75.86±0.05 | 52.18±0.35 | 42.03±0.90 | 44.72±1.03 | 24.38±1.02 | 25.00±2.04 | 21.71±3.52 |
| | node2vec [15] | 75.53±0.01 | 76.00±0.01 | 75.92±0.06 | 52.64±0.62 | 41.42±0.99 | 44.14±0.89 | 23.84±2.31 | 23.40±2.65 | 22.83±1.69 |
| | GAE [23] | 80.54±0.43 | 72.55±0.52 | 76.60±1.11 | 58.33±0.46 | 42.25±0.87 | 40.89±1.09 | 62.45±0.44 | 66.11±0.49 | 61.54±0.53 |
| | GCN [22] | 80.93±0.19 | 73.53±1.93 | 78.10±0.41 | 54.77±0.73 | 54.17±0.70 | 42.45±0.97 | 69.05±0.86 | 74.53±0.36 | 70.58±0.68 |
| | GAT [49] | 79.59±0.61 | 65.64±0.33 | 74.91±1.31 | 54.84±0.37 | 39.63±0.16 | 53.28±0.78 | 53.80±1.53 | 55.85±1.62 | 50.37±1.72 |
| | GIN [56] | 75.70±0.57 | 73.11±0.11 | 74.90±0.17 | 52.39±0.31 | 44.48±0.84 | 58.39±0.23 | 59.10±0.18 | 66.27±0.27 | 60.46±0.25 |
| Source-Free | SHOT [25] | 80.63±0.11 | 75.23±0.33 | 76.20±0.21 | 56.94±0.27 | 50.94±0.07 | 52.62±0.79 | 73.32±0.48 | 74.16±1.88 | 69.81±1.08 |
| | SHOT++ [27] | 80.80±0.06 | 74.69±0.33 | 76.27±0.38 | 56.57±0.29 | 52.04±0.56 | 49.97±0.48 | 71.51±0.93 | 74.99±0.90 | 70.73±0.59 |
| | BNM [6] | 80.80±0.08 | 74.56±0.41 | 76.48±0.04 | 57.92±0.16 | 51.39±0.22 | 50.78±1.13 | 73.59±0.31 | 75.83±0.64 | 69.96±0.42 |
| | ATDOC [26] | 80.39±0.32 | 74.43±0.50 | 76.40±0.20 | 56.31±0.44 | 49.02±0.58 | 42.65±0.16 | 72.01±0.35 | 74.80±0.45 | 67.64±1.44 |
| | NRC [57] | 80.79±0.19 | 74.09±1.26 | 75.24±0.38 | 56.96±0.41 | 50.63±0.09 | 50.83±0.46 | 70.89±0.39 | 71.79±0.34 | 68.44±0.86 |
| | DaC [67] | 80.11±0.18 | 76.17±0.33 | 78.47±0.41 | 58.09±0.55 | 55.97±0.97 | 56.55±0.30 | 73.02±0.51 | 74.75±0.93 | 68.81±0.47 |
| | JMDS [24] | 82.92±0.25 | 76.29±0.36 | 79.69±0.31 | 56.67±0.20 | 48.72±0.08 | 46.93±0.26 | 68.28±1.13 | 72.68±0.47 | 64.96±0.63 |
| | GTrans [20] | 81.93±0.29 | 75.66±0.46 | 78.97±0.10 | 56.35±0.15 | 61.30±0.17 | 60.80±0.26 | 64.85±0.99 | 71.44±1.65 | 67.27±0.25 |
| | SOGA [34] | 82.81±0.18 | 76.32±0.33 | 78.97±0.41 | 58.27±0.60 | 53.71±0.32 | 57.14±0.49 | 71.62±0.37 | 74.16±0.72 | 67.06±0.32 |
| Ours | GraphCTA | **85.82±0.88** | **79.47±0.35** | **81.23±0.61** | **59.85±0.16** | **63.35±0.84** | **63.18±0.31** | **75.62±0.29** | **77.62±0.22** | **72.56±0.43** |

**Table 3: Performance with different components.**

| Models | A→D | C→D | C→A |
|---|---|---|---|
| SOGA [34] | 71.62 | 74.16 | 67.06 |
| Source Pretrained Model (SPM) | 65.07 | 70.12 | 61.88 |
| SPM + $\mathcal{L}_M$ (Model Adaptation) | 73.32 | 75.31 | 71.05 |
| SPM + $\mathcal{L}_G$ (Graph Adaptation) | 66.47 | 73.92 | 64.13 |
| GraphCTA | **75.62** | **77.62** | **72.56** |

**Table 4: Performance with different graph adaptation strategies.**

| Models | A→D | C→D | C→A |
|---|---|---|---|
| SPM | 65.07±0.12 | 70.12±0.25 | 61.88±0.09 |
| SUBLIME [29] | 65.75±0.12 | 67.37±0.26 | 68.69±0.57 |
| SLAPS [11] | 65.99±0.84 | 72.77±0.73 | 67.54±0.91 |
| SPM + $\mathcal{L}_G$ | 66.47±0.04 | 73.92±0.14 | 64.13±0.21 |
| GraphCTA | **75.62±0.29** | **77.62±0.22** | **72.56±0.43** |

**Table 5: Combine graph adaptation with other models.**

| Architectures | A→D | C→D | C→A |
|---|---|---|---|
| SHOT | 73.32±0.48 | 74.16±1.88 | 69.81±1.08 |
| SHOT + $\mathcal{L}_G$ | 67.39±0.10 | 76.86±0.08 | 69.62±0.03 |
| BNM | 73.59±0.31 | 75.83±0.64 | 69.96±0.42 |
| BNM + $\mathcal{L}_G$ | 62.12±0.98 | 67.22±0.95 | 69.58±0.05 |
| GraphCTA | **75.62±0.29** | **77.62±0.22** | **72.56±0.43** |

tasks. The performance lift can be attributed to the collaborative mechanism between model adaptation and graph adaptation. The presented results demonstrate its effectiveness in facilitating source-free unsupervised graph domain adaptation.

## 4.3 Ablation Study

In this subsection, we conduct ablation studies on citation datasets and similar conclusions can be drawn from the remaining datasets. *Comprehensive parameter sensitivity analyses are provided in Appendix F for further details.*

*4.3.1 The Effect of Model Adaptation and Graph Adaptation.* To investigate the contribution of model adaptation and graph adaptation in GraphCTA, we show the effectiveness of our proposed collaborative mechanism in Table 3. Specifically, the source-pretrained model is denoted as SPM and we strength the SPM with model adaptation ($\mathcal{L}_M$) and graph adaptation ($\mathcal{L}_G$), respectively. As we can see, both two modules improve the performance of SPM, but the model adaptation module plays a more significant role compared with the graph adaptation module. This is because the model often captures more generic or transferable knowledge across domains, while graph adaptation might be less crucial when the underlying structures or relationships in the graphs are already aligned. In comparison, our method incorporates these two modules into a

collaborative paradigm and surpasses all alternatives by a significant margin. Note that our GraphCTA, even with model adaptation alone, surpasses the performance of SOGA, which serves as additional evidence of the effectiveness of our GraphCTA.

*4.3.2 The Alternative Graph Adaptation Strategies.* As we have discussed in Section 3.3, there exist lots of choices to perform graph adaptation. Here, we present two additional graph structure learning strategies to conduct graph adaptation. While graph structure learning has been extensively studied in the literature [11, 13, 29], most existing methods depend highly on node labels, which are not available in our unsupervised graph domain adaptation setting. To this end, we choose two recent unsupervised graph structure learning models SUBLIME [29] and SLAPS [11] to refine the graph structure, where both of them utilize self-supervised learning techniques. Among them, SUBLIME [29] employs GNN to learn node similarity matrix and KNN-based sparsification is used to produce

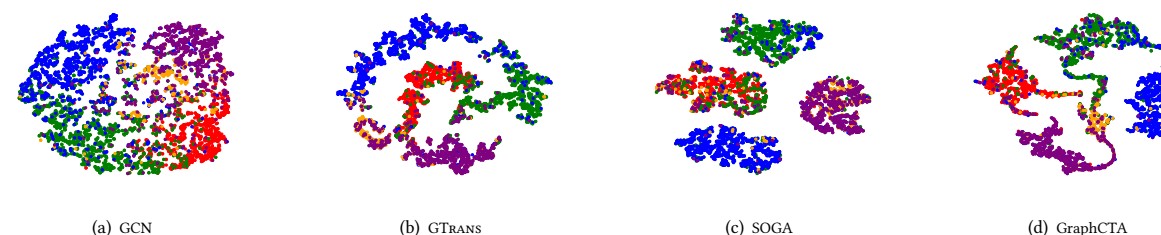

(a) GCN        (b) GTRANS        (c) SOGA        (d) GraphCTA

**Figure 2: Visualizations of target graph node representations with each color representing a class in citation networks (C→D).**

**Table 6: Results with different architectures.**

| Architectures | A→D | C→D | C→A |
|---|---|---|---|
| GraphCTA$_{GCN}$ | **75.62±0.29** | **77.62±0.22** | **72.56±0.43** |
| GraphCTA$_{GAT}$ | 71.84±0.52 | 72.04±0.87 | 66.91±0.82 |
| GraphCTA$_{SAGE}$ | 73.50±0.41 | 73.65±0.26 | 68.17±0.34 |
| GraphCTA$_{GIN}$ | 72.92±0.39 | 73.85±0.54 | 71.26±0.17 |

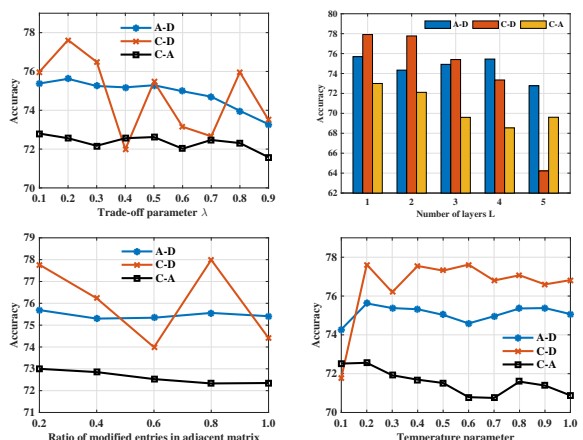

**Figure 3: Hyper-parameter sensitivity analysis.**

sparse adjacent matrix. Similarly, SLAPS [11] utilizes a denoising autoencoder loss as self-supervision. Table 4 demonstrates the performance with different graph adaptation strategies. As we can see, SUBLIME and SLAPS are not as good as our strategy except in the scenario of C→A. Notably, SUBLIME occasionally exhibits inferior performance compared to the source-pretrained model (SPM), particularly due to its reliance on data augmentation operations. In contrast, our strategy does not require such operations and exhibits high versatility. Furthermore, we also explore the integration of our graph adaptation strategy with various existing model adaptation approaches, as shown in Table 5. Surprisingly, a simple combination of these two modules often leads to a decline in performance. It becomes evident that a collaborative approach is necessary to achieve optimal results, thus emphasizing the novelty and effectiveness of our proposed GraphCTA method.

*4.3.3 Architectures and Hyper-parameter Analyses.* As discussed in previous section, our proposed GraphCTA is model-agnostic and can be integrated into various GNN architectures. We investigate the impacts of 4 widely used GNN backbones: GCN [22], GAT [49], GraphSAGE [16] and GIN [56]. Their results are showed in Table

6. In general, the performance varies across different graph neural network architectures, which is also influenced by the used datasets. We observe that the GAT architecture performs worst, since the learned attention weights in source graph are not suitable in target graph and it has more parameters to be fine-tuned due to the multi-head attention mechanism. The simplest GCN architecture surprisingly works well in most cases. At last, we further show the impacts of several hyper-parameters in Figure 3. Particularly, when setting $\lambda = 0.2$, $L = 1$ or 2, budget $\mathcal{B} = 0.2|\mathbf{A}|$ and $\tau = 0.2$, our model could obtain the best performance. *More hyper-parameter analyses are provided in Appendix F.*

### 4.4 Visualization

To gain an intuitive understanding of the learned node representations, we use t-SNE [48] to project the node representations into a 2-D space. Figure 2 presents the scatter plots generated by GCN [22], GTRANS [20], SOGA [34] and our proposed GraphCTA from $C \rightarrow D$, where each color represents a distinct class. It can be observed that the vanilla GCN without adaptation operations fails to produce satisfactory results, as nodes from different classes are mixed together. While two representative source-free baselines GTRANS and SOGA are capable of clustering nodes together, the boundary between these clusters are quite blurred, resulting in only four clusters with significant overlapping. In contrast, our proposed GraphCTA demonstrates the ability to learn more compact node representations within the same category. This highlights its effectiveness in learning discriminative node representations even in the presence of domain shift.

### 5 CONCLUSION

We investigate a relatively unexplored area in graph representation learning, i.e., source-free unsupervised graph domain adaptation, where the labelled source graph is not available due to privacy concerns. Specifically, we propose a novel framework named GraphCTA that performs model adaptation and graph adaptation collaboratively to mitigate the source hypothesis bias and domain shift. The whole framework is model-agnostic and optimized via an alternative strategy. We conduct comprehensive experiments on three public datasets with fifteen adaptation tasks, which demonstrates the effectiveness of our proposed model compared with recent state-of-the-art baselines. In the future, it would be an intriguing challenge to explore how to extend the GraphCTA framework to handle more domain adaptation tasks, such as source-free semi-supervised graph DA and source-free open-set graph DA.

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

## A  THEORETICAL DETAILS

Shen et al. [40, 66] provides a generalization bound on domain adaptation through applying Wasserstein distance [2] between source and target domain distributions. For completeness, we present Definition A.1 and Theorem A.2 as follows:

*Definition A.1 (Wasserstein Distance).* The $\rho$-th Wasserstein distance between two distributions $\mathbb{D}_S$ and $\mathbb{D}_T$ is defined as follow:

$$\mathcal{W}_\rho(\mathbb{D}_S, \mathbb{D}_T) = (\inf_{\gamma \in \Pi[\mathbb{D}_S, \mathbb{D}_T]} \iint d(x_s, x_t)^\rho d\gamma(x_s, x_t))^{1/\rho}, \quad (13)$$

where $\Pi[\mathbb{D}_S, \mathbb{D}_T]$ is the set of all joint distribution on $\mathcal{X} \times \mathcal{X}$ with marginals $\mathbb{D}_S$ and $\mathbb{D}_T$. $d(x_s, x_t)$ is a distance function for two instances $x_s, x_t$.

THEOREM A.2. *Given two domain distributions $\mathbb{D}_S$ and $\mathbb{D}_T$, denote $f^* = \arg\min_{f \in \mathcal{H}}(\epsilon_T(f) + \epsilon_S(f))$ and $\xi = \epsilon_T(f^*) + \epsilon_S(f^*)$. Assume all hypotheses $h$ are $K$-Lipschitz continuous, the risk of hypothesis $\hat{f}$ on the target domain is then bounded by:*

$$\epsilon_T(\hat{f}) \le \epsilon_S(\hat{f}) + 2K\mathcal{W}(\mathbb{D}_S, \mathbb{D}_T) + \xi, \quad (14)$$

*where $\mathcal{W}_1$ distance is used and we ignore the subscript 1 for simplicity.*

With the above definition and theorem, we can know that the target domain prediction error is bounded by summarizing the source domain prediction error, the distribution divergence of source and target domains, and the combined error $\xi$. Most existing domain adaptation methods can be regarded as minimizing the distribution divergence [42, 52, 60], i.e., the second term in Eq. (14). However, in the source-free setting, the source data are inaccessible, hence the right part is not applicable. Therefore, we need a new generation upper bound for source-free target domain adaptation.

Our proposed GraphCTA mainly consists of two key modules: model adaptation and graph adaptation, where the objective functions are designed to constrain the upper bound. Specifically, we utilize structural neighborhood consistency to provide guiding information in model adaptation module. That's to say, the source distribution $\mathbb{D}_S$ is replaced with $\mathbb{D}_N = \bigcup_{x \in \mathbb{D}_T} \mathcal{B}(x, r)$, where $\mathcal{B}(x, r) = \{x' : x' \in A_x \wedge \|x' - x\| \le r\}$, where $\|\cdot\|$ is $L_1$ distance function, $r > 0$ is a small radius, and $A_x$ is sample $x's$ neighborhood. With a small $r$, we have $\mathcal{W}(\mathbb{D}_N, \mathbb{D}_T) = \inf_{\gamma \in \Pi[\mathbb{D}_N, \mathbb{D}_T]} \iint \|x_n - x_t\| d\gamma(x_n, x_t) \le r$, where for each $x_t \in \mathbb{D}_T$ we can find at least one $x_n \in \mathbb{D}_N$ such that $\|x_n - x_t\| \le r$. Thus, the overall distance will be bounded by $r$ and the domain divergence is reduced. Furthermore, the graph adaptation module aims to correct the covariate-shift in the input space and [20, 67] have prove its capability in reducing prediction error. Then, we have the following Theorem A.3:

THEOREM A.3. *Given domain distribution $\mathbb{D}_T$ and $\mathbb{D}_N$, where $\mathbb{D}_N = \bigcup_{x \in \mathbb{D}_T} \mathcal{B}(x, r)$ and $\mathcal{B}(x, r) = \{x' : x' \in A_x \wedge \|x' - x\| \le r\}$ provide guiding information through local neighborhood. Denote $f^* = \arg\min_{f \in \mathcal{H}}(\epsilon_T(f) + \epsilon_N(f))$ and $\xi = \epsilon_T(f^*) + \epsilon_N(f^*)$. Assume that all hypotheses $h$ are $K$-Lipschitz continuous, the risk of hypothesis $\hat{f}$ on the target domain is then bounded by:*

$$\epsilon_T(\hat{f}) \le \epsilon_N(\hat{f}) + 2Kr + \xi, \quad (15)$$

*where a small $r$ will reduce the bound.*

Thus, it can be inferred that the joint application of model adaptation and graph adaptation can lead to a reduction in the terms on the right-hand side of Eq. (15), resulting in the minimization of the upper bound for the prediction error on the target domain.

## B  OPTIMIZATION

The optimization process for GNN parameters and $\Delta \mathbf{X}$ is straightforward as they can be updated using gradient descent due to their differentiability. However, optimizing $\Delta \mathbf{A}$ is notably challenging because of its binary nature and constrained properties. Therefore, we relax $\Delta \mathbf{A}$ to continuous space $[0, 1]^{n \times n}$ and utilize projected gradient descent (PGD) [14, 55] to update its elements:

$$\Delta \mathbf{A} \leftarrow \Pi_{\mathcal{B}}(\Delta \mathbf{A} - \eta \frac{\partial \mathcal{L}_G}{\partial \Delta \mathbf{A}}), \quad (16)$$

where the gradient step is performed with step size $\eta$, and then it is projected into the constrained space $\mathcal{B}$. We further constrain the search space of $\Delta \mathbf{A}$ to the existing edges in graph. More specifically, $\Pi_{\mathcal{B}}(\cdot)$ is expressed as:

$$\Pi_{\mathcal{B}}(\mathbf{x}) = \begin{cases} \Pi_{[0,1]}(\mathbf{x}), & \text{if } \mathbf{1}^\top \Pi_{[0,1]}(\mathbf{x}) \le \mathcal{B}, \\ \Pi_{[0,1]}(\mathbf{x} - \gamma \mathbf{1}) \text{ s.t. } \mathbf{1}^\top \Pi_{[0,1]}(\mathbf{x} - \gamma \mathbf{1}) = \mathcal{B}. \end{cases} \quad (17)$$

where $\Pi_{[0,1]}(\cdot)$ restricts the input values to the range $[0, 1]$. $\mathbf{1}$ represents a vector with all elements equal to one, and $\gamma$ is determined by solving the equation $\mathbf{1}^\top \Pi_{[0,1]}(\mathbf{x} - \gamma \mathbf{1}) = \mathcal{B}$ with the bisection method [28]. To keep sparsity, we regard each entry as a Bernoulli distribution and sample the learned graph structure as $\mathbf{A}' \sim \text{Bernoulli}(\mathbf{A} \oplus \Delta \mathbf{A})$.

## C  TRAINING STRATEGY FOR GRAPHCTA

We outline the training strategy for GraphCTA and present the pseudo codes in Algorithm 1.

## D  DATASETS AND BASELINES

### D.1  Distribution Shifts on Graphs

We choose three widely used node classification datasets for domain adaptation, i.e., transaction, social and citation graphs. These datasets contain a varying number of nodes, ranging from thousands to tens of thousands. The statistical information of these datasets is presented in Table 1. In each dataset, they share the same input feature space and output label space, but the characteristics from different graphs often exhibit distinct properties, which results in domain shift. Here, we utilize degree assortativity [1] and clustering coefficient [39] as measures to describe the structural properties of these graphs. Specifically, a high assortativity score indicates that nodes with high degrees are more likely to connect with other high degree nodes, while the clustering coefficient measures the extent to which nodes in a graph tend to form tightly clusters. We provide a quantitative comparison in Figure 4. As we can see, these graphs demonstrate significant disparities in their statistics, suggesting the presence of distribution shifts w.r.t. graph structures.

---

**Algorithm 1** GraphCTA's Training Strategy

---

**Input:** Given source pretrained GNN model $\kappa = f_\theta \circ g_\phi$ and target graph $\mathcal{G} = (\mathbf{A}, \mathbf{X})$

**Output:** Predictions $\mathbf{Y} \in \mathbb{R}^{n \times C}$ on refined target graph $\mathcal{G} = (\mathbf{A}', \mathbf{X}')$ with updated model $\kappa'$

1: $\Delta\mathbf{X}$ and $\Delta\mathbf{A}$ are initialized as zero matrices

2: **while** not converged or not reached the maximum epochs **do**

3:     Compute $\mathbf{X}' = \sigma(\mathbf{X}) = \mathbf{X} + \Delta\mathbf{X}$ and $\mathbf{A}' = \psi(\mathbf{A}) = \mathbf{A} \oplus \Delta\mathbf{A}$

4:     **for** $i \leftarrow 1$ to $T_m$ **do**                                        ▷ model adaptation

5:         Fix the parameters of $\Delta\mathbf{X}$ and $\Delta\mathbf{A}$

6:         Compute node representations $\mathbf{Z} \in \mathbb{R}^{n \times h}$ and predictions $\mathbf{P} \in \mathbb{R}^{n \times C}$ with $(\mathbf{A}', \mathbf{X}')$

7:         Calculate $\mathcal{L}_M$ according to Eq. (8) and update GNN model's parameters

8:         Update memory banks $\mathcal{F}$ and $\mathcal{P}$ via a momentum manner

9:     **for** $j \leftarrow 1$ to $T_f$ **do**                          ▷ graph adaptation for node features

10:         Fix the parameters of GNN model and $\Delta\mathbf{A}$

11:         Calculate $\mathcal{L}_G$ according to Eq. (12) and update $\Delta\mathbf{X} \leftarrow \Delta\mathbf{X} - \eta \frac{\partial \mathcal{L}_G}{\partial \Delta\mathbf{X}}$

12:     **for** $k \leftarrow 1$ to $T_s$ **do**                            ▷ graph adaptation for structure

13:         Fix the parameters of GNN model and $\Delta\mathbf{X}$

14:         Calculate $\mathcal{L}_G$ according to Eq. (12) and update $\Delta\mathbf{A} \leftarrow \Pi_{\mathcal{B}}(\Delta\mathbf{A} - \eta \frac{\partial \mathcal{L}_G}{\partial \Delta\mathbf{A}})$ with PGD

15: Update target graph as $\mathbf{X}' = \sigma(\mathbf{X}) = \mathbf{X} + \Delta\mathbf{X}$ and $\mathbf{A}' = \psi(\mathbf{A}) = \text{Bernoulli}(\mathbf{A} \oplus \Delta\mathbf{A})$

16: Compute predictions $\mathbf{Y} = \kappa'(\mathbf{A}', \mathbf{X}')$ with updated model $\kappa'$

---

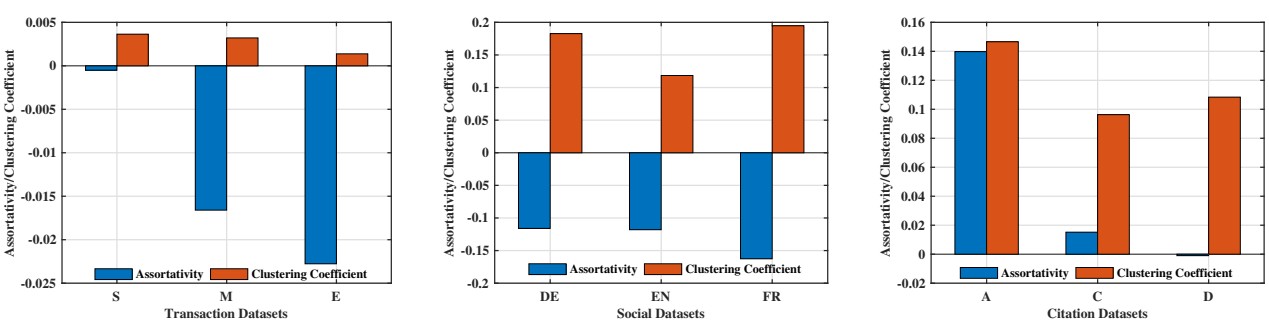

Figure 4: Graph structure properties statistic, which shows the existence of distribution shifts.

## D.2 Baseline Settings

In our experiments, we compare our proposed GraphCTA with 3 groups of approaches: *no-adaptation*, *source-need* and *source-free*. The detailed description is as follows:

**No-adaptation.** This group contains 3 unsupervised algorithms (i.e., DeepWalk [37], node2vec [15], GAE [23]) and 3 graph neural networks including GCN [22], GAT [49] and GIN [56]. For unsupervised methods, we first learn node representations in source graph via a unsupervised manner. Then, the source label information is utilized to train a logistic regression classifier on the learned source node representations. After that, we learn node representations in target graph via a unsupervised manner and evaluate its performance with the logistic regression classifier trained on source data. For GNN models, they are directly trained with the labelled source graph and evaluated on the target graph, because they can be optimized in an end-to-end manner.

**Source-need.** Methods of this group utilize the labelled source graph to eliminate the distribution discrepancies explicitly or implicitly. Among them, UDAGCN [53], AdaGCN [7], ASN [64] and ACDNE [41] employ adversarial training to implicitly minimize the distribution divergence. In contrast, TPN [35], GRADE [52]

and SpecReg [60] explicitly regularize the node representations via spectral or MMD loss to reduce the domain shifts. Since the labelled source graph can provide supervision signals, they are able to achieve relatively better performance compared with source-free methods.

**Source-free.** In this group, we consider several recent state-of-the-art source-free baselines. Since there is limited research exploring source-free unsupervised graph domain adaptation, we adapt several baselines from the field of computer vision. Specifically, SHOT [25] and its extension SHOT++ [27] employ entropy minimization and information maximization to perform class-wise adaptation. BNM [6] achieves prediction discriminability and diversity via nuclear norm maximization. ATDOC [26] and NRC [57] exploit local neighborhood structure for ensuring label consistency. DaC [67] partitions the target data into source-like and target-specific samples to perform domain adaptation. JMDS [24] assign confidence score to each target sample for robust adaptation learning. GTRANS [20] performs graph transformation at test time to enhance the model's performance. SOGA [34] maximizes the mutual information between the target graph and the output of the model to preserve the structural proximity.

**Table 7: Average node classification performance in terms of accuracy with standard deviation (%) on _social datasets_. We use blue to denote the best performance in source-need methods and bold indicates the best performance among source-free approaches.**

| | Methods | EN→DE | FR→DE | FR→EN | DE→EN | DE→FR | EN→FR |
|---|---|---|---|---|---|---|---|
| Source-Need | UDAGCN [53] | 58.69±0.75 | 63.11±0.44 | 55.11±0.22 | 59.74±0.21 | 56.61±0.39 | 56.94±0.70 |
| | TPN [35] | 53.82±1.26 | 43.72±0.83 | 46.41±0.19 | 54.42±0.19 | 43.43±0.99 | 38.93±0.28 |
| | AdaGCN [7] | 51.31±0.68 | 42.15±0.21 | 47.04±0.12 | 54.69±0.50 | 37.62±0.51 | 40.45±0.24 |
| | ASN [64] | 60.45±0.16 | 39.54±0.63 | 45.43±0.88 | 55.45±0.11 | 47.20±0.84 | 40.29±0.55 |
| | ACDNE [41] | 58.79±0.73 | 55.14±0.43 | 54.50±0.45 | 58.08±0.97 | 54.01±0.30 | 57.15±0.61 |
| | GRADE [52] | 61.18±0.08 | 52.02±0.14 | 49.74±0.05 | 56.40±0.05 | 46.83±0.07 | 51.17±0.62 |
| | SpecReg [60] | 61.45±0.18 | 61.97±0.21 | 56.29±0.42 | 56.43±0.11 | 63.20±0.03 | 63.21±0.04 |
| No-Adaptation | DeepWalk [37] | 55.08±0.61 | 41.67±0.93 | 46.84±0.99 | 52.18±0.35 | 42.03±0.90 | 44.72±1.03 |
| | node2vec [15] | 54.61±1.53 | 41.42±0.83 | 46.83±0.54 | 52.64±0.62 | 41.42±0.99 | 44.14±0.89 |
| | GAE [23] | 54.57±0.49 | 44.49±1.31 | 48.06±0.81 | 58.33±0.46 | 42.25±0.87 | 40.89±1.09 |
| | GCN [22] | 52.02±0.17 | 45.37±1.46 | 47.32±0.33 | 54.77±0.73 | 54.17±0.70 | 42.45±0.97 |
| | GAT [49] | 43.65±0.37 | 43.76±0.74 | 45.52±0.13 | 54.84±0.37 | 39.63±0.16 | 53.28±0.78 |
| | GIN [56] | 55.26±0.75 | 55.67±0.75 | 54.18±0.09 | 52.39±0.31 | 44.48±0.84 | 58.39±0.23 |
| Source-Free | SHOT [25] | 58.95±0.40 | 61.26±0.26 | 56.40±0.11 | 56.94±0.27 | 50.94±0.07 | 52.62±0.79 |
| | SHOT++ [27] | 63.61±0.20 | 61.01±0.59 | 55.12±0.41 | 56.57±0.29 | 52.04±0.56 | 49.97±0.48 |
| | BNM [6] | 61.83±0.24 | 60.94±0.31 | 56.70±0.20 | 57.92±0.16 | 51.39±0.22 | 50.78±1.13 |
| | ATDOC [26] | 61.95±0.28 | 57.47±0.89 | 54.22±0.43 | 56.31±0.44 | 49.02±0.58 | 42.65±0.16 |
| | NRC [57] | 63.08±0.34 | 61.84±0.34 | 56.12±0.65 | 56.96±0.41 | 50.63±0.09 | 50.83±0.46 |
| | DaC [67] | 62.58±0.34 | 55.61±0.77 | 57.73±0.48 | 58.09±0.55 | 55.97±0.97 | 56.55±0.30 |
| | JMDS [24] | 61.48±0.08 | 62.12±0.14 | 52.35±0.32 | 56.67±0.20 | 48.72±0.08 | 46.93±0.26 |
| | GTRANS [20] | 62.00±0.17 | 62.06±0.23 | 56.54±0.06 | 56.35±0.15 | 61.30±0.17 | 60.80±0.26 |
| | SOGA [34] | 62.55±1.38 | 50.22±0.58 | 50.11±0.23 | 58.27±0.60 | 53.71±0.32 | 57.14±0.49 |
| Ours | GraphCTA | **63.85±0.83** | **62.45±0.23** | **58.39±0.41** | **59.85±0.16** | **63.35±0.84** | **63.18±0.31** |

**Table 8: Average node classification performance in terms of accuracy with standard deviation (%) on _citation datasets_. We use blue to denote the best performance in source-need methods and bold indicates the best performance among source-free approaches.**

| | Methods | A→D | C→D | D→A | C→A | A→C | D→C |
|---|---|---|---|---|---|---|---|
| Source-Need | UDAGCN [53] | 66.95±0.45 | 71.77±1.09 | 58.16±0.19 | 66.80±0.23 | 72.15±0.92 | 73.28±0.52 |
| | TPN [35] | 69.78±0.69 | 74.65±0.74 | 62.99±1.25 | 67.93±0.34 | 74.56±0.73 | 72.54±1.08 |
| | AdaGCN [7] | 75.04±0.49 | 75.59±0.71 | 69.67±0.54 | 71.67±0.91 | 79.32±0.85 | 78.20±0.90 |
| | ASN [64] | 73.80±0.40 | 76.36±0.33 | 70.15±0.60 | 72.74±0.49 | 80.64±0.27 | 78.23±0.52 |
| | ACDNE [41] | 76.24±0.53 | 77.21±0.23 | 71.29±0.66 | 73.59±0.34 | 81.75±0.29 | 80.14±0.09 |
| | GRADE [52] | 68.22±0.37 | 73.95±0.49 | 63.72±0.88 | 69.55±0.78 | 76.04±0.57 | 74.32±0.54 |
| | SpecReg [60] | 75.93±0.89 | 75.74±1.15 | 71.01±0.64 | 72.04±0.63 | 80.55±0.70 | 79.04±0.83 |
| No-Adaptation | DeepWalk [37] | 24.38±1.02 | 25.00±2.04 | 23.88±4.27 | 21.71±3.52 | 23.63±2.37 | 23.70±2.96 |
| | node2vec [15] | 23.84±2.31 | 23.40±2.65 | 23.47±2.92 | 22.83±1.69 | 23.37±3.72 | 23.56±3.68 |
| | GAE [23] | 62.45±0.44 | 66.11±0.49 | 52.79±1.30 | 61.54±0.53 | 64.98±0.53 | 60.53±0.87 |
| | GCN [22] | 69.05±0.86 | 74.53±0.36 | 63.35±0.69 | 70.58±0.68 | 77.38±1.28 | 74.17±1.15 |
| | GAT [49] | 53.80±1.53 | 55.85±1.62 | 52.93±1.84 | 50.37±1.72 | 57.13±1.73 | 55.52±1.78 |
| | GIN [56] | 59.10±0.18 | 66.27±0.27 | 58.98±0.28 | 60.46±0.25 | 68.61±0.36 | 69.25±0.34 |
| Source-Free | SHOT [25] | 73.32±0.48 | 74.16±1.88 | 62.86±1.73 | 69.81±1.08 | 76.81±1.41 | 74.94±1.65 |
| | SHOT++ [27] | 71.51±0.93 | 74.99±0.90 | 65.50±0.64 | 70.73±0.59 | 76.77±0.74 | 76.70±1.05 |
| | BNM [6] | 73.59±0.31 | 75.83±0.64 | 65.83±0.67 | 69.96±0.42 | 78.91±0.34 | 76.87±0.75 |
| | ATDOC [26] | 72.01±0.35 | 74.80±0.45 | 63.67±0.88 | 67.64±1.44 | 76.94±0.92 | 74.89±0.99 |
| | NRC [57] | 70.89±0.39 | 71.79±0.34 | 65.25±0.56 | 68.44±0.86 | 75.93±0.70 | 76.19±0.66 |
| | DaC [67] | 73.02±0.51 | 74.75±0.93 | 65.18±1.87 | 68.81±0.47 | 77.43±0.70 | 76.78±0.72 |
| | JMDS [24] | 68.28±1.13 | 72.68±0.47 | 59.41±1.32 | 64.96±0.63 | 70.84±1.27 | 70.40±0.53 |
| | GTRANS [20] | 64.85±0.99 | 71.44±1.65 | 63.47±1.93 | 67.27±0.25 | 69.05±0.34 | 72.27±0.29 |
| | SOGA [34] | 71.62±0.37 | 74.16±0.72 | 66.00±0.35 | 67.06±0.32 | 77.05±0.56 | 75.53±0.94 |
| Ours | GraphCTA | **75.62±0.29** | **77.62±0.22** | **70.04±0.15** | **72.56±0.43** | **80.55±0.13** | **79.56±0.27** |

# E    MORE EXPERIMENTAL RESULTS

Table 7 and Table 8 present all adaptation results on social and citation datasets. As transaction datasets exhibit temporal shifts, we only focus on performing adaptation tasks that involve transitioning from previous graphs to later graphs (i.e., S→M, S→E and E→M in Table 2). From Table 7 and Table 8, we have the following observations: (1) The no-adaptation baselines perform poorly in most scenarios, since they do not take target graph into consideration thus failing to model the domain shifts. (2) In general, the source-need methods could achieve relatively good performance, because the labelled source graph provides available supervision

signals to directly minimize their distribution discrepancies. (3) While the source-free setting is challenging, we can still obtain satisfied results via employing appropriate learning paradigms. Our proposed model gains significant improvements over recent SOTA baselines, which verifies the effectiveness of GraphCTA.

## F   MORE HYPER-PARAMETER ANALYSES

We provide more hyper-parameter sensitivity analyses in Figure 5. Specifically, we explore the impact of various key hyperparameters by varying them across different scales. For the trade-off parameter $\lambda$, the model's performance gradually decreases when its value exceeds a certain threshold (i.e., 0.2). This indicates weighted cross-entropy loss plays a more important role compared with contrastive loss, as it provides explicit supervision signals. Similar trends are also identified in parameters $\alpha$ and $\beta$. In particular, we note that it exhibits significant oscillation during the adaptation task from $C$ to $D$ across all three parameters. This maybe because the training procedure is sensitive to the incorporation of contrastive learning module in this particular adaptation scenario. On the other hand, GraphCTA is relatively robust to the number of nearest neighbors $K$ and temperature $\tau$. When the number of $K$ increases, there is a slight decline in its performance, which might be caused by the introduced noisy neighbors. Nevertheless, a smaller value can consistently yield better performance.

## G   EXPERIMENTAL RESULTS REPRODUCE

We present the detailed running configurations for all the compared methods. As the methods in [6, 24–27, 57, 67] are specifically designed for image data, their codes cannot directly run on the graph-structured data. Thus, we replace their backbones with the same GCN [22] architecture used in our model for fair comparisons. We conduct our experiments on a Linux server with a NVIDIA's A100 GPU. The embedding size is set to 128 for each method. The code sources and other specific hyper-parameter settings of compared methods are listed as below.

**DeepWalk** [37] and **node2vec** [15]. We use the codes provided by Pytorch Geometric. The walk length is set as 20 and window size is set to 10. We set the number of walks for each node to 10 and the number of negative samples for each training pair is set to 1. For node2vec, we set parameters $p = 0.5$ and $q = 2$.

**GAE** [23], **GCN** [22], **GAT** [49] and **GIN** [56]. We also use the implementation in Pytorch Geometric. The number of layers and node representations are set as 2 and 128, respectively. The learning rate and weight are search in the range of $\{0.1, 0.01, 1e^{-3}, 1e^{-4}, 5e^{-4}\}$.

For the remaining baselines, we use the source codes provided by the authors at Github if available. Their links are as follows:

- **UDAGCN** [53]: https://github.com/GRAND-Lab/UDAGCN
- **AdaGCN** [7]: https://github.com/daiquanyu/AdaGCN
- **ASN** [64]: https://github.com/yuntaodu/ASN
- **ACDNE** [41]: https://github.com/shenxiaocam/ACDNE
- **GRADE** [52]: https://github.com/jwu4sml/GRADE
- **SpecReg** [60]: https://github.com/Shen-Lab/GDA-SpecReg
- **SHOT** [25]: https://github.com/tim-learn/SHOT
- **SHOT++** [27]: https://github.com/tim-learn/SHOT-plus
- **BNM** [6]: https://github.com/cuishuhao/BNM
- **ATDOC** [26]: https://github.com/tim-learn/ATDOC

**Table 9: Citation datasets used in SpecReg and SOGA.**

| Dataset | #Nodes | #Edges | #Features | #Classes |
|---|---|---|---|---|
| DBLPv8 | 5,578 | 7,341 | 7,537 | 6 |
| ACMv9 | 7,410 | 11,135 | | |

*Two adaptation tasks: D→A and A→D.

**Table 10: Citation datasets used in our paper.**

| Dataset | #Nodes | #Edges | #Features | #Classes |
|---|---|---|---|---|
| ACMv9 | 9,360 | 15,556 | | |
| Citationv1 | 8,935 | 15,098 | 6,775 | 5 |
| DBLPv7 | 5,484 | 8,117 | | |

*Six adaptation tasks: D→A, A→D, C→A, A→C, C→D, D→C.

- **NRC** [57]: https://github.com/Albert0147/NRC_SFDA
- **DaC** [67]: https://github.com/ZyeZhang/DaC
- **JMDS** [24]: https://github.com/Jhyun17/CoWA-JMDS
- **GTRANS** [20]: https://github.com/ChandlerBang/GTrans

As **SOGA** [34] do not release its source codes, we try our best to implement it based on the descriptions in its paper. Our proposed **GraphCTA** is implemented with Pytorch Geometric [12] and optimized with Adam optimizer [21]. The optimal learning rate and weight decay are searched in $\{0.1, 0.01, 1e^{-3}, 1e^{-4}, 5e^{-4}\}$. The smoothing parameter $\gamma$ in memory banks is fixed as 0.9 by default. Temperature $\tau$ and the number of $K$-nearest neighbors are set as 0.2 and 5, respectively. Trade-off hyper-parameters $\lambda, \alpha, \beta$ are searched in the range of $[0, 1]$.

## H   ADDITIONAL EXPERIMENTS

### H.1   Comparisons on SpecReg/SOGA datasets

We note that two recent baselines SpecReg [60] and SOGA [34] utilize the citation datasets that are similar yet distinct from the citation datasets used in our paper. SpecReg and SOGA follow UDAGCN [53], which provides 2 domains with node feature dimension of 7,537 and number of classes as 6. (Note that UDAGCN also uses Citationv2 dataset in their paper, but they do not release this dataset.) Our paper utilizes the widely used citation datasets provided by AdaGCN [7], ASN [64] and ACDNE [41], which provides 3 domains with node features dimension of 6,775 and number of classes as 5. This dataset provides us with the opportunity to explore a broader range of adaptation settings, encompassing six adaptation tasks instead of two. The detailed statistical information is summarized in Table 9 and Table 10. As can be seen, their datasets are different from ours in number of nodes, edges, node features and number of classes.

To show the effectiveness of our proposed GraphCTA, we provide the results on same datasets used by SpecReg and SOGA in Table 11. For source-need baseline SpecReg, we report the results in their original papers and the results reproduced by their released source codes. As for source-free baseline SOGA, the authors do not release their source codes and we have tried our best to reproduce their results. As we can see in the table, our reproduced SOGA results are very closed to their reported results, which means our reproduced codes for SOGA are reliable. Moreover, it is worth noting that our proposed source-free GraphCTA outperforms SOGA with different gains and achieves comparable performance with source-need baseline SpecReg.

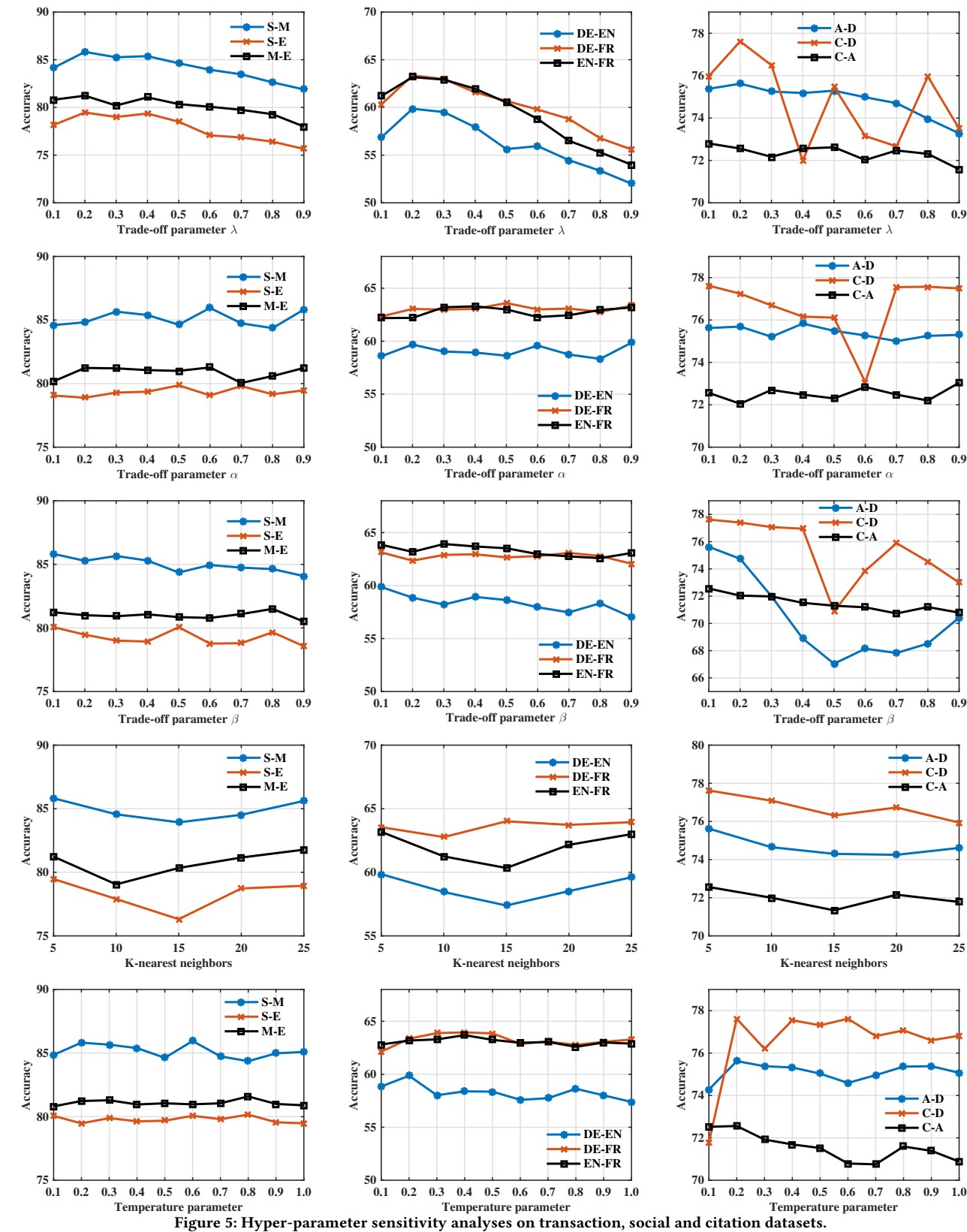

**Figure 5: Hyper-parameter sensitivity analyses on transaction, social and citation datasets.**

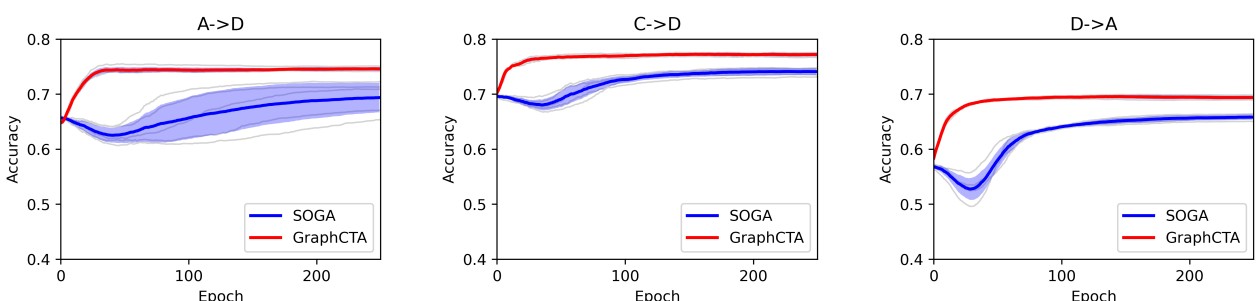

Figure 6: The comparison of learning curves between GraphCTA and SOGA.

Table 11: Results on citation datasets used by SpecReg and SOGA.

| Methods | DBLPv8→ACMv9 | | ACMv9→DBLPv8 | |
|---|---|---|---|---|
| | Macro-F1 | Micro-F1 | Macro-F1 | Micro-F1 |
| SpecReg (report) | - | 76.26±0.05 | - | 91.65±0.06 |
| SpecReg (reproduce) | 65.83±0.28 | 75.52±0.17 | 91.96±0.74 | 91.30±0.80 |
| SOGA (report) | 63.60±0.30 | - | 92.80±1.80 | - |
| SOGA (reproduce) | 63.47±1.32 | 71.94±1.15 | 91.35±1.82 | 91.22±1.90 |
| GraphCTA | 64.58±0.72 | 74.08±0.15 | 91.95±0.29 | 91.62±0.52 |

## H.2 Comparisons on training convergence

We compare the training convergence of GraphCTA and SOGA in Figure 6. Each model is trained 5 times with random seeds (i.e., 1,2,3,4,5). The light gray lines are the results for each experiment. We plot the mean accuracy curve and fill the area within its standard deviation. As can be seen, our proposed GraphCTA converges with fewer epochs and is more stable with smaller deviations.

