# OpenReview forum: "Collaborate to Adapt: Source-Free Graph Domain Adaptation via Bi-directional Adaptation"
_ACM.org/TheWebConf/2024/Conference — TheWebConf24_

### Official Review · Reviewer_rjLH · 2023-10-24

**Novelty:** 4
**Technical Quality:** 4

**Review:**

The paper introduces a new framework named GraphCTA, designed for tackling the complex challenges of source-free graph domain adaptation. By ingeniously combining model and graph adaptation, the framework addresses the distribution shifts and source hypothesis bias commonly observed in graph-structured data. GraphCTA employs a dual adaptation strategy: on the model side, it uses local neighborhood predictions and global class prototypes for adaptation, while on the graph side, it leverages predictions made by the model along with stored data in memory banks for further refinement. Extensive experiments validate the utility of GraphCTA, showcasing its superior performance over existing state-of-the-art baselines across multiple scenarios.

**Questions:**

1. While the paper introduces GraphCTA as a novel framework, its technical contributions, specifically the use of pseudo-labeling and contrastive learning, are well-established techniques in the field. Therefore, the innovation in methodology could be considered incremental. What is the own advantage of your methods? More detail is better.

2. In the introduction, the paper could modify its claim to better align with the mention of related work later in the text. Instead of stating that "there has been limited investigation of source-free adaptation techniques for the non-iid graph-structured data," it could say: "While there has been some work (add cite), which focuses on source-free unsupervised graph domain adaptation, the field remains relatively under-explored." This adjustment would provide a more accurate picture of the current state of research and seamlessly connect with the discussion of SOGA in the related work section.

3. While the paper claims the novelty of collaboratively bi-directional adaptations as a significant contribution, the absence of experimental results focusing solely on one type of adaptation limits the ability to measure the effectiveness of this collaborative approach. Moreover, the paper could benefit from additional experiments specifically designed to validate the claimed synergistic effects of model and graph adaptation working in tandem.

**Reviewer Confidence:**

4: The reviewer is certain that the evaluation is correct and very familiar with the relevant literature

**Scope:**

4: The work is relevant to the Web and to the track, and is of broad interest to the community

---

### Official Review · Reviewer_ZFxg · 2023-11-13

**Novelty:** 5
**Technical Quality:** 5

**Review:**

Strength

The paper addresses a significant challenge in graph domain adaptation: adapting a pre-trained model to a target graph without access to the labeled source graph. This is particularly relevant in scenarios where data privacy and regulation constraints make source data inaccessible

Weakness
1. The representation of the paper is bad. First, the domain is not cleared defined in the beginning which causes difficulty in reading the introduction part. Fsecond, there is almost no explaination for Figure 1, which makes the figure useless.

2. The experiment didn't show what if the graph is used for a higher level domain adaptation like transfer from different datasets.

3. Adversarial attack is also an important domain shifting. Experiments should be provided to compare the robustness of their method against adversarial attack.

4. For different methods, the backbone training on source domain should use the same training procedure or even the same pre-trained model. The author use the code of different adaption method, which may cause some unfair comparison. The accuracy of source graph should be reported to make sure the performance of proposed method is not from the extensive hyperpamameter tuning on source graph.

**Questions:**

What is the performance on the source domain? That is important for comparison. Since the author didn’t use the same code, the performance on source domain may be different. But they should be the same for fairness.

**Reviewer Confidence:**

4: The reviewer is certain that the evaluation is correct and very familiar with the relevant literature

**Scope:**

4: The work is relevant to the Web and to the track, and is of broad interest to the community

---

### Official Review · Reviewer_nwhM · 2023-11-21

**Novelty:** 5
**Technical Quality:** 6

**Review:**

The paper addresses the graph domain adaptation problem in source-free unsupervised setting. The novel paradigm, GraphCTA is introduced, which follows a series of procedures: model adaptation, graph adaptation, and model adaptation with updated graph as new input. This can be viwed as a collaborative loop between model adaptation and graph adaptation. Extensive experiments supports the efficacy of Graph CTA.


- The paper addresses an interesting problem: GDA in unsupervised setting.
- The paper is well organized, and easy to follow. The theorems are stated clearly, and proven nicely.
- Extensive experiments are conducted. Source code is provided.

**Questions:**

Q1. While extensive experiments are conducted, only accuracy is used as main evaluation metric. Is there any reason F1 was not used for evaluation?

**Reviewer Confidence:**

2: The reviewer is willing to defend the evaluation, but it is likely that the reviewer did not understand parts of the paper

**Scope:**

4: The work is relevant to the Web and to the track, and is of broad interest to the community

---

### Official Review · Reviewer_sGRw · 2023-11-25

**Novelty:** 5
**Technical Quality:** 6

**Review:**

In this paper, authors propose a novel framework for source-free graph domain adaptation. They conduct model adaptation and graph adaptation collaboratively with different training process. Complexity analysis is also provided for all modules. Experiments on different datasets could show the superiority of their model on the source-free scenario. Ablation experiments could prove the effectiveness of all components in their framework.

The strong points of this work could be summarized as follows:

1.The problem they study is meaningful.

2.The model they propose is novel enough.

3.The experiments on three datasets among different settings could show the superiority of their method.

4.The parameter analysis and ablation experiments could prove the effectiveness of all components in their framework.

However, there are some drawbacks as:

1.The variable should be defined before utilization. For example, the m and z_i in Eq(2) are not defined, which make this work hard to be understood.

2.The scales of the datasets they use are not large enough.

3.The performance improvement is tiny.

**Questions:**

Could this model be applied on large dataset?

**Reviewer Confidence:**

4: The reviewer is certain that the evaluation is correct and very familiar with the relevant literature

**Scope:**

3: The work is somewhat relevant to the Web and to the track, and is of narrow interest to a sub-community

---

### Decision · Program_Chairs · 2024-01-22

**Decision:**

Accept

**Comment:**

**Meta-review**: The paper addresses the graph domain adaptation problem in a *source-free*, unsupervised setting. Unsupervised GDA is an interesting problem and reviewers generally liked the paper. The discussion was productive and improved the paper.

 **Strengths**:
 + problem setting; unsupervised GDA (nwhM, sGRw, ZFxg, rjLH)
 + paper organization (nwhM)
 + experiments (nwhM, sGRw)

 **Weaknesses**:
 - relatively small performance improvement (sGRw)